# Reactive Paper Spray Ionization Mass Spectrometry for Rapid Detection of Estrogens in Cosmetics

**DOI:** 10.3390/molecules28155675

**Published:** 2023-07-27

**Authors:** Dongning Song, Jing Liu, Yang Liu

**Affiliations:** 1National Institutes for Food and Drug Control, Beijing 102629, China; 18251981389@163.com (D.S.); liujing_zsm@126.com (J.L.); 2NMPA Key Laboratory for Quality Research and Evaluation of Chemical Drugs, Beijing 102629, China

**Keywords:** RPSI, estrogens, cosmetics, online derivatization

## Abstract

Rapid detection of harmful estrogens in cosmetics is essential in protecting public health. To reduce time-consuming pretreatment and analytical procedures, a novel reactive paper spray ionization mass spectrometry (RPSI-MS) methodology was developed. RPSI-MS is suitable for quantitatively analyzing estrogens in cosmetics by utilizing an online derivatization reaction between estrogens and 2-fluoro-1-methyl-pyridinium-p-toluene-sulfonate (FluMP). Using estradiol valerate as the internal standard (I.S.), three estrogens, estradiol, estriol, and ethinyloestradiol, in cosmetics were quantitatively characterized within minutes. Multiple parameters were optimized including FluMP concentration and volume, triethylamine amount as well as the drying time. The three estrogens displayed good linearity ranging from 0.002 to 1 μg/mL, with R^2^ above 0.99. The recovery results of all the estrogens were within 80~111%. The limit of detection (LOD) was 0.001 μg/mL for the three estrogens. Compared to conventional paper spray ionization mass spectrometry (PSI-MS), extraction is not required and the detection sensitivity of RPSI-MS was improved by 34,000, 80,000, and 1400 times for estradiol, estriol, and ethinyloestradiol, respectively. The protocol established in this paper is sensitive, eco-friendly, and suitable for rapid testing of estrogens in cosmetics.

## 1. Introduction

Hormones are vital information-transferring molecules that communicate between cells to benefit the cell population [1]. Hormones have several significant effects on the human body, such as regulating gene expression to exert potent influences on reproductive physiology [2], regulating nociceptive pathway [3], etc. Sex hormones are one type of hormone that are classified into three classes: estrogens, androgens, and progestogens. Even at low concentrations, these compounds play a crucial role in numerous physiological functions, including reproduction, sexual differentiation, pregnancy, regulation of the immune system [4,5], and displaying “neuroprotective” activity in the brain [6,7]. Estrogens may have physiological effects such as making skin white, eliminating wrinkles, preventing skin from aging, and are thus illicitly added to cosmetics [8]. Prolonged use of cosmetics containing estrogens can lead to various side effects, including metabolic dysfunctions [8], breast cancer, and endometrial hyperplasia [8,9]. The European Union’s Cosmetics Regulations [Regulation (EC) No 1223/2009] [10] have categorically prohibited the use of estrogens in cosmetics. Thus, it is essential to develop simple and effective methods for the detection and quantification of estrogens present in cosmetic products.

Various methods have been developed for detecting estrogens in cosmetics. De-Zhu Tu used second-order calibration coupled with excitation-emission matrix fluorescence to detect estriol and estrone in liquid cosmetics [8]. Jean C. Hubinger employed high-performance liquid chromatography with ultraviolet detection (HPLC-UV) to identify estriol, estradiol, estrone, and progesterone in cosmetic products [11]. Daniela De Orsi used a high-performance liquid-chromatography–diode array and electrospray mass spectrometry to analysis non-allowed substances in cosmetic products for preventing hair loss and other hormone-dependent skin diseases [12]. However, these methods are time-consuming and require complex pretreatment, which highlights the significance of developing straightforward, effective methodologies to overcome these challenges.

Mass spectrometry (MS) is an analytical technique that can provide qualitative and quantitative information on molecules. However, conventional MS requires complex preprocessing to avoid contamination. Ambient ionization mass spectrometry (AIMS), which was first introduced in 2004 by Cooks et al. [13], overcomes this difficulty and allows for operation in an atmosphere environment with minimal pretreatment. Paper spray ionization mass spectrometry (PSI-MS) is a type of AIMS [14] that combines the advantages of AIMS and electrospray ionization (ESI).

The recently developed reactive paper spray ionization mass spectrometry (RPSI-MS) combines the features of PSI-MS and online derivatization to enable rapid qualitative and quantitative analysis of compounds with low mass spectrum responses in complex mixtures without extensive preprocessing. RPSI-MS has been employed to detect various compounds, including quinones [15], aldehydes [16], and to assess the chemical reactivity of the Katritzky reaction [17].

Here, we developed a RPSI-MS method for rapidly detecting estradiol, estriol, and ethinyloestradiol (Figure 1) in cosmetics utilizing FluMP as the derivatization reagent. This protocol requires almost no pretreatment and can quantitatively determine the content of estrogens in cosmetics within minutes.

## 2. Results and Discussion

### 2.1. The Requirements of Online Derivatization

Online derivatization offers several advantages, including improved sensitivity, convenience, and rapid analysis. Additionally, it is more environmentally friendly than traditional solution-phase reactions. The selection of an appropriate derivatization reagent is critical to ensure proper reaction. Factors to consider for RPSI-MS include the speed of the reaction, formation of one or more positive charges in the product to improve sensitivity, and that the derivative reaction does not interfere with other materials present in cosmetics. It is reported that RPSI-MS accelerates the derivatization reaction during the formation of electrospray and ionization, which is much more efficient than the corresponding bulk solution-phase reactions performed on the same scale in acetonitrile solvent [18,19,20].

FluMP was selected as the derivatization reagent to react with estrogens due to its ability to react with primary and secondary alcohols, as well as phenols in complex mixtures [21,22]. Previously, our study showed that FluMP can effectively react with estrogens in solution (Figure 2) [23]. FluMP was also used to react and improve the signal response of cannabinoids [24] and buprenorphine [25]. After derivatization, a quaternary ammonium salt was introduced to the target estrogens and this significantly improved the signal response.

### 2.2. The Optimization of FluMP’s Concentration and Volume

To ensure complete reaction during derivatization, an appropriate amount of FluMP must be used. However, excessive amounts of FluMP can result in the blockage of the paper’s tip. Therefore, after optimization, 5 μL of 1 mg/mL FluMP was used for the derivatization process.

### 2.3. Examination of the Addition Position and Amount of Triethylamine

Triethylamine may offer basic conditions and is essential to the derivatization reaction. Experiments showed that pre-mix trimethylamine with the derivatization reagent and pre-mix trimethylamine with spray solvent, or pre-loading trimethylamine on the paper base yielded poor signals during MS tests. Meanwhile, adding triethylamine to the estrogen solution (*v*/*v* = 1:10) and directly loading the resulting solution onto the paper generated the highest response signals.

### 2.4. Optimize the Time of the Product Loaded on the Paper

The adsorption and desorption of the reagents and products on the paper were affected by the loading and drying time. After loading, the concentration of reactants increases with the evaporation of solvents, which will drive the reaction in the positive direction. However, the increase in drying time may also cause precipitation and inhibit desorption. The results indicated that as the loading and drying time increased, the signal response did not increase significantly. In this study, the spray solvent was applied directly after loading the sample onto the paper and yielded good results.

### 2.5. Linearity, Lower Limits of Detection

Mass Hunter (Agilent, Palo Alto, CA, USA) was used for quantification. Calibration curves were created by plotting the response of the analyte to IS against nominal concentrations of analyte standards to establish linearity (Figure 3 and Table 1). The lower limits of detection (LOD) were determined using a signal-to-noise ratio (S/N) of three.

### 2.6. Recovery

Recovery experiments were performed to assess the accuracy of the method. A blank sample of 50 mg was mixed with 500 μL of linear standard solution with different concentrations (0.01 μg/mL, 0.1 μg/mL, 1 μg/mL), followed by the addition of 5 μL of internal standard solution and 50 μL of triethylamine. The resulting solution was vortexed for 10 s and then analyzed using the RPSI-MS method. Three replicates were performed per concentration, and their relative standard deviations (RSD) were all less than 10%. The average recoveries were determined as follows: estradiol—105.8% (0.01 μg/mL), 95.3% (0.1 μg/mL), 85.7% (1 μg/mL); estriol—110.0% (0.01 μg/mL), 97.0% (0.1 μg/mL), 83.0% (1 μg/mL); ethinyloestradiol—84.0% (0.01 μg/mL), 80.5% (0.1 μg/mL), 85.0% (1 μg/mL).

### 2.7. Detection Sample with Complex Matrix

The practicability of the method was verified by reinforcing the quantified target compounds into a complex matrix consisting of water, butanediol, glyceryl polyacrylate, glycerinum, β-glucan, etc., and then determining their content using this method. The results indicated that this method could accurately and rapidly quantify the amount of added estrogens in cosmetics (see Table 2).

### 2.8. Future Applications

For molecules that only contain carbon, hydrogen, and oxygen atoms, the ionization efficacy is poor and maybe not suitable for paper spray ionization. RPSI-MS is an effective method for the rapid analysis of these type of compounds such as estrogens, vitamins, and lipids. The quick reactions and high sensitivity can help the drug cosmetics quality control. Since different compounds have various functional groups, there is not a universal derivatization reagent available for compounds containing different functional groups. Finding eligible derivatization reagents is still interesting yet challenging.

## 3. Materials and Methods

### 3.1. Reagents and Materials

Estradiol, estriol, and ethinyloestradiol were from the National Institutes for Food and Drug Control (Beijing, China). Acetonitrile was purchased from Sigma-Aldrich (St. Louis, MO, USA). The 2-fluoro-1-methyl-pyridinium-p-toluene-sulfonate was purchased from TCI chemicals (Shanghai, China). Triethylamine was from HUSHI (Shanghai, China). All the reagents were used directly without further purification.

### 3.2. Instrument

All experiments were carried out with an Agilent 1290 HPLC coupled with a 6470 triple quadruple mass spectrometer (Palo Alto, CA, USA). Data were acquired and processed by Agilent MassHunter Workstation 10.1 (Palo Alto, CA, USA). HB-Z303-1AC High-voltage DC Power supply (Tianjin, China) and KQ-500DA CNC ultrasonic cleaner (Kunshan, China) were used. Grade 1 chromatographic paper was from Whatman (Stevenage, UK).

### 3.3. Sample Preparation

A mixed standard stock solution of 10 μg/mL containing approximately 0.2 mg each of estradiol, estriol, and ethinyloestradiol in 20 mL acetonitrile was prepared. Linear standard stock solutions were prepared by further diluting the mixed standard solution to 0.002, 0.01, 0.1, 0.5, and 1 μg/mL with acetonitrile. An internal standard solution of 200 μg/mL estradiol valerate was also prepared by transferring 2 mg estradiol valerate to a 10 mL volumetric flask and diluting to volume with acetonitrile. For preparing the linear standard solution, 5 μL of internal standard solution was added to 500 μL of the linear standard solution, followed by the addition of 50 μL triethylamine and mixing. Sample solutions were prepared by dissolving 50 mg of the sample in 500 μL ACN, then 5 μL of the internal standard and 50 μL of trimethylamine were added in sequence. Finally, the resulting solutions were mixed.

### 3.4. Mass Spectrometry Parameters

The filter paper used in this experiment was Whatman grade 1 paper and was cut into isosceles triangles measuring 5 mm × 13 mm. A spray solvent consisting of 20 μL acetonitrile:H_2_O = 9:1 was used, and the spray voltage was set to 3.5 kV. The distance between the paper tip and the mass spectrometer cone was approximately 0.5 cm. All three estrogens were detected in positive ion mode with parent ions and daughter ions of 364.2→128.1, 380.2→128.1, and 388.2→128.1, respectively. A collision energy of 65 V was used for all three compounds. Both the gas and sheath temperatures were maintained at 100 °C. Other parameters were automatically optimized by the instrument.

### 3.5. RPSI-MS Analysis

To perform the analysis, 5 μL of a 1 mg/mL derivatization reagent solution was spotted and dried onto a triangular paper. Next, 5 μL of either the standard or sample solution was added to the paper. Once the solution was added, a direct current power supply was applied, and 20 μL of spray solvent was added to initiate spray desorption. The signals were then collected (Figure 4).

## 4. Conclusions

In this study, a reactive paper spray mass spectrometry (RPSI-MS) method was proposed to analyze estradiol, estriol, and ethinyloestradiol in liquid cosmetic samples. Compared to normal PSI, the method greatly improved sensitivity, with estradiol, estriol, and ethinyloestradiol sensitivity increasing by 34,000, 80,000, and 1400 times, respectively. RPSI-MS is suitable for compounds that have a low response in MS, and through derivatization, significant improvements can be achieved. In the future, more universal derivative reagents may be studied, and the method could also be applied to characterize estrogens in blood or urine. In summary, RPSI-MS is a new method that holds great potential for detecting compounds with low response in traditional PSI-MS.

## Figures and Tables

**Figure 1 molecules-28-05675-f001:**
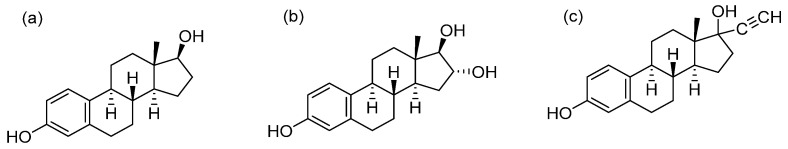
Structures of estradiol (**a**), estriol (**b**), and ethinyloestradiol (**c**).

**Figure 2 molecules-28-05675-f002:**
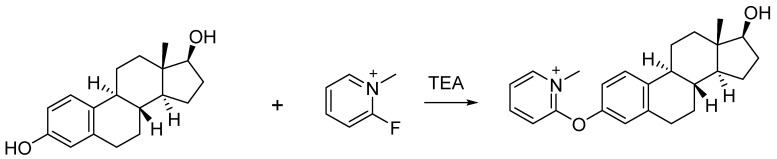
The reaction between FluMP and estradiol.

**Figure 3 molecules-28-05675-f003:**
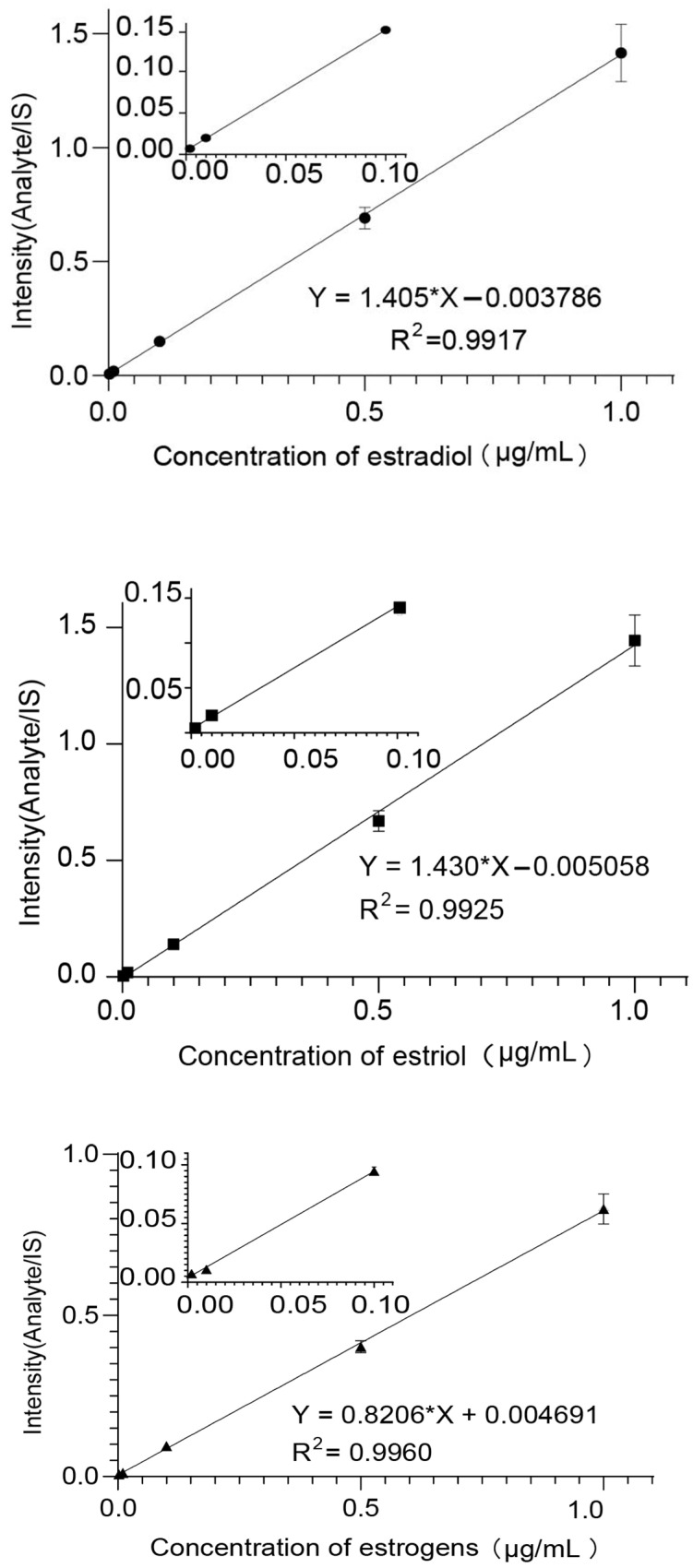
The linear curves of estradiol, estriol, and ethinyloestradiol.

**Figure 4 molecules-28-05675-f004:**
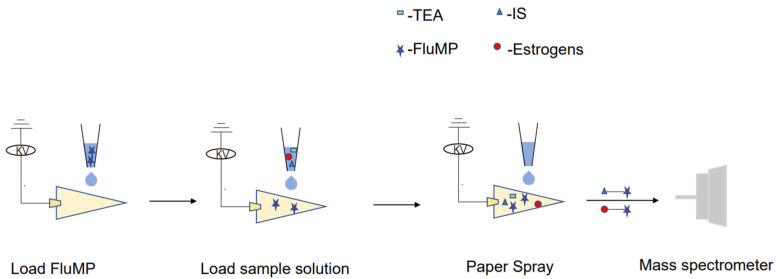
A schematic diagram of the analysis of the sample using the RPSI-MS system.

**Table 1 molecules-28-05675-t001:** The results of linearity and lower limits of detection.

Compound	Linearity Range	Linear Regression	LOD (μg/mL)	R^2^
Estradiol	0.002~1 μg/mL	Y = 1.405x − 0.003786	0.001	0.9917
Estriol	0.002~1 μg/mL	Y = 1.430x − 0.005058	0.001	0.9925
Ethinyloestradiol	0.002~1 μg/mL	Y = 0.8206x − 0.004691	0.001	0.9960

**Table 2 molecules-28-05675-t002:** The results of detection samples with the complex matrix.

Estradiol			Estriol			Ethinyloestradiol		
Addition (μg/g, *n* = 3)	Detection (μg/g, *n* = 3)	RSD (%, *n* = 3)	Addition (μg/g, *n* = 3)	Detection (μg/g, *n* = 3)	RSD (%, *n* = 3)	Addition (μg/g, *n* = 3)	Detection (μg/g, *n* = 3)	RSD (%, *n* = 3)
0.1	0.11	5.9	0.1	0.11	5.1	0.1	0.08	3.6
1	0.95	9.0	1	0.97	9.8	1	0.80	2.1
10	8.57	2.3	10	8.30	8.0	10	8.50	5.1

## Data Availability

The data presented in this study are available on request from the corresponding author.

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
