# Peer review of "Reactive Paper Spray Ionization Mass Spectrometry for Rapid Detection of Estrogens in Cosmetics"

_molecules, 2023, doi:10.3390/molecules28155675_

Round 1

Reviewer 1 Report

 Minor revision:

-In abstract Lines 15-16, The three estrogen displayed good linearity ranging from 0.002 to 1 μg/mL, with R² above 0.99. The dynamic range is quite high. By RPSI-MS, it is questionable. The authors are requested to check it and that can reduce as logical.

-In line 17, The limit of detection (LOD) was 0.001 μg/mL, however, the authors claimed that the dynamic range of detection was from 0.002 to 1 μg/mL. It must be clarified.

-In line 81, Not “--- our studied showed----“, it must be “--- our study showed----“,

-In Table 2, it is better to use ‘n’ instead of ‘N’,

-What is the reason for observing recovery rate more than 100%?,  

-In line 164, acetonitrile:H2O=9:1, please check subscript,

-In line 167, A collision energy of 65 eV was used for------------, if it is CID, then % instead of eV is the right unit,

In Fig 4, “Figure 4. The schematic diagram of detection the sample” or Figure 4. A schematic diagram of analysis of the sample using RPSI-MS system”, please check it,

-In lines 178-179, “------------ proposed to measure estradiol, estriol, and ethinyloestradiol---------“, please rephrase this sentence as ““------------ proposed to analyze estradiol, estriol, and ethinyloestradiol---------“,

-In line 185, “In summary, RPSI-MS is a new technology method that holds great potential for------“,

-Please check typos and grammatical mistakes.

Author Response

Dear Editor, Reviewer:

Thank you very much for your valuable comments. We read the comments carefully and made changes according to the suggestions in the manuscript. All the changes are labelled in the manuscript for clear reading. Below are the replies:

Reviewer1

  1. In abstract Lines 15-16, The three estrogen displayed good linearity ranging from 0.002 to 1 μg/mL, with R² above 0.99. The dynamic range is quite high. By RPSI-MS, it is questionable. The authors are requested to check it and that can reduce as logical.

Thanks for the reviewer’s suggestions. Experimental data showed good linearity ranging from 0.002 to 1 μg/mL. To address reviewer’s question, we updated and added linearity ranges from 0.002 to 0.1μg/mL in Figure 3.

2 -In line 17, The limit of detection (LOD) was 0.001 μg/mL, however, the authors claimed that the dynamic range of detection was from 0.002 to 1 μg/mL. It must be clarified.

The limits of detection (LOD) was estimated by S/N = 3, where S represents the signal in the TIC chromatogram and N represents the noise of the TIC chromatogram. At the concentration of 0.002 μg/mL, the S/N was approximate 6, thus the LOD was 0.001 μg/mL for the three estrogens.

3 -In line 81, Not “--- our studied showed----“, it must be “--- our study showed----“,

Thanks, the words have been changed.

4 -In Table 2, it is better to use ‘n’ instead of ‘N’,

Letters have been changed.

5 -What is the reason for observing recovery rate more than 100%?

Theoretically, the recovery rate should be not more than 100%. But due to systematic error and uncertainty, the calculated recovery rate may be higher than 100%. According to ISO 17025, at 10 ppm concentration, the accepted recovery rate is 80 – 115%.

6 -In line 164, acetonitrile:H2O=9:1, please check subscript,

Thanks, subscript was applied.

7 -In line 167, A collision energy of 65 eV was used for------------, if it is CID, then % instead of eV is the right unit,

Thanks for the comment. Different instrument vendors may have different expression in collision energy. For Agilent mass spectrometer, the collision energy unit was V, and it was corrected in the article.

8 -In Fig 4, “Figure 4. The schematic diagram of detection the sample” or Figure 4. A schematic diagram of analysis of the sample using RPSI-MS system”, please check it,

The sentence has been updated.

9 -In lines 178-179, “------------ proposed to measure estradiol, estriol, and ethinyloestradiol---------“, please rephrase this sentence as ““------------ proposed to analyze estradiol, estriol, and ethinyloestradiol---------“,

The sentence has been rephrased, thanks.

10 -In line 185, “In summary, RPSI-MS is a new technology method that holds great potential for------“,

Thanks, the sentence has been changed.

11 -Please check typos and grammatical mistakes.

Thanks for the reviewer’s comments. We checked and corrected the grammatical mistakes.

Reviewer 2 Report

I have the following comments:

1.      The aim is not clearly defined in the abstract.

2.      Keywords are way too long.

3.      The aim should be written at the end of the introduction part.

4.      Figure 1 and 2 should be cited.

5.      It should be better explained why triethylamine was tested.

6.      Table 2: standard deviations should be included.

7.      The authors should include more references and improve the results and discussion part.

Author Response

Dear Editor, Reviewer:

Thank you very much for your valuable comments. We read the comments carefully and made changes according to the suggestions in the manuscript. All the changes are labelled in the manuscript for clear reading. Below are the replies:

Reviewer2

  1. The aim is not clearly defined in the abstract.

Thanks for the valuable comments. The abstract has been rephrased.

  1. Keywords are way too long.

The keywords have been modified.

  1. The aim should be written at the end of the introduction part.

Thanks for the reviewer’s suggestions. The aim has been added at the end of the introduction part.

  1. Figure 1 and 2 should be cited.

Figure 1 and 2 have be cited, thanks.

  1. It should be better explained why triethylamine was tested.

Our study found that the amount of triethylamine is essential for the derivatization reaction. We also found that pre-mix triethylamine with the derivatization reagent or pre-loading trimethylamine on the paper base greatly affect the reaction yield.

Why trimethylamine was tested was updated in the manuscript.

  1. Table 2: standard deviations should be included.

Standard deviations has been added in Table 2.

  1. The authors should include more references and improve the results and discussion part

More references about mechanisms and influencing factors of on-line derivatization have been added (ref. 18 - 20). The results and discussion parts were improved, thanks.
